# Deep Learning-Based Computer-Aided Diagnosis of Rheumatoid Arthritis with Hand X-ray Images Conforming to Modified Total Sharp/van der Heijde Score

**DOI:** 10.3390/biomedicines10061355

**Published:** 2022-06-08

**Authors:** Hao-Jan Wang, Chi-Ping Su, Chien-Chih Lai, Wun-Rong Chen, Chi Chen, Liang-Ying Ho, Woei-Chyn Chu, Chung-Yueh Lien

**Affiliations:** 1Department of Information Management, National Taipei University of Nursing and Health Sciences, Taipei City 112, Taiwan; majoajoa@gmail.com (H.-J.W.); cpsu00@outlook.com (C.-P.S.); keiraaa06@gmail.com (W.-R.C.); pdil728@gmail.com (C.C.); elain890706@gmail.com (L.-Y.H.); 2Division of Allergy, Immunology, and Rheumatology, Department of Medicine, Taipei Veterans General Hospital, Taipei City 112, Taiwan; cclai3@vghtpe.gov.tw; 3Institute of Clinical Medicine, School of Medicine, National Yang Ming Chao Tung University, Taipei City 112, Taiwan; 4Faculty of Medicine, School of Medicine, National Yang Ming Chiao Tung University, Taipei City 112, Taiwan; 5Department of Biomedical Engineering, National Yang Ming Chiao Tung University, Taipei City 112, Taiwan; wchu@nycu.edu.tw

**Keywords:** wrist joint detection, joint space narrowing, erosion, mTSS, artificial intelligence

## Abstract

Introduction: Rheumatoid arthritis (RA) is a systemic autoimmune disease; early diagnosis and treatment are crucial for its management. Currently, the modified total Sharp score (mTSS) is widely used as a scoring system for RA. The standard screening process for assessing mTSS is tedious and time-consuming. Therefore, developing an efficient mTSS automatic localization and classification system is of urgent need for RA diagnosis. Current research mostly focuses on the classification of finger joints. Due to the insufficient detection ability of the carpal part, these methods cannot cover all the diagnostic needs of mTSS. Method: We propose not only an automatic label system leveraging the You Only Look Once (YOLO) model to detect the regions of joints of the two hands in hand X-ray images for preprocessing of joint space narrowing in mTSS, but also a joint classification model depending on the severity of the mTSS-based disease. In the image processing of the data, the window level is used to simulate the processing method of the clinician, the training data of the different carpal and finger bones of human vision are separated and integrated, and the resolution is increased or decreased to observe the changes in the accuracy of the model. Results: Integrated data proved to be beneficial. The mean average precision of the proposed model in joint detection of joint space narrowing reached 0.92, and the precision, recall, and F1 score all reached 0.94 to 0.95. For the joint classification, the average accuracy was 0.88, and the accuracy of severe, mild, and healthy reached 0.91, 0.79, and 0.9, respectively. Conclusions: The proposed model is feasible and efficient. It could be helpful for subsequent research on computer-aided diagnosis in RA. We suggest that applying the one-hand X-ray imaging protocol can improve the accuracy of mTSS classification model in determining mild disease if it is used in clinical practice.

## 1. Introduction

Rheumatoid arthritis (RA) is a systemic autoimmune disease that usually affects multiple joints in the hands, wrists, and knees. Joints with RA become inflamed over time, causing damage to connective tissue. According to studies, an average of 460 per 100,000 people worldwide will have RA between 1980 and 2019, with a predicted range of 95% (0.06–1.27) and a total prevalence of approximately 1–2% of the global population [1]. The conventional diagnosis of RA relies on hand and foot X-rays and musculoskeletal ultrasound images. Currently, the mTSS (the modified total Sharp score) is widely used to diagnose the severity of RA on hand and foot radiographs to define different levels of joint erosion and joint space narrowing (JSN). This scale has recently become a gold standard for evaluating the efficacy of new medications for RA and dry hand arthritis, for reducing and discontinuing medications for disease recurrence, and for changing treatment strategies [2].

In clinical practice, the mTSS scoring process requires evaluating every joint of both fingers and toes to label the total number of joints. Accordingly, is a time-consuming and labor-intensive process, which is the top barrier to its adoption into daily clinical practice. Although the mTSS is a very useful tool aiding decision making in diagnosis and treatment for RA patients, the lack of efficient and time-reserving tools for diagnosis limits the widespread use of mTSS in clinical practice.

For precision medicine, the mTSS is also a very important evaluation factor to evaluate the use of biopharmacy for RA treatment. The dosages and formulations of biopharmacological drugs need to be precise, so that the effectiveness of the treatment of RA can be effectively improved. For example, Tanaka et al. used the mTSS to discontinue the systemic immunosuppressive agent ADA plus methotrexate when the degree of degeneration showed clinical remission (i.e., ΔmTSS ≤ 0.5) to avoid toxicity and side-effects caused by prolonged administration. Lin et al. found that RA patients with sarcopenia had a longer disease duration and higher disease activity, which could be assessed by mTSS scores, and the degree of degeneration was exacerbated in overweight patients with a high risk of degenerative progression and sarcopenia, and in patients with both overweight and sarcopenia [3].

The mTSS plays an important role in both medication decisions and inhibiting RA progression. The precise use of the mTSS in conjunction with other tools, e.g., Simple Disease Activity Index (SDAI), Low Disease Activity (LDA), and Health Assessment Questionnaire Disability Index (HAQ-DI), is a major factor for assessing the degree of degradation of the RA. Those tools can be combined with the mTSS to assist immuno-rheumatologists to set treatment strategies to achieve early prevention and maintain degenerative stability, avoid bone destruction and disability, and attain a treat-to-target (T2T) approach [4].

The development of a rapid mTSS scoring system combined with a clinical decision-making system could provide immuno-rheumatologists with the possibility of establishing treatment strategies for RA, as well as a diagnostic support for medication decisions. Eventually, this might facilitate clinical decisions to use alternative drugs with fewer side-effects and lower toxicity, as well as reduce the use of more toxic drugs, thereby improving the quality of life of patients, relieving their suffering, and prolonging the progression of RA patients.

### 1.1. AI in Imaging in RA

Existing studies on automatic radiographic scoring of RA patients mainly focus on detecting joint regions in ultrasound images and the joint space measurement of X-ray images. For example, Hemalatha et al. used a CNN model to detect ultrasound images for calculating the radiographic score and reached an average accuracy of 90% [5]. Norgeot et al. proposed a longitudinal DL model with electronic health records to predict the development of patients with RA, and the overall model performance measured by the AUC-ROC reached 0.91 [6]. Dang and Allison also used a CNN model to score the RA severity level in X-ray images, including 16 joints in the left and right hands and six joints in the left and right feet, with an average accuracy of 0.908 [7]. In addition, Rohrbach et al. proposed a CNN model to assist physicians in scoring the RA severity level [8]. They mentioned that there was no significant difference between their model and the evaluation results of human experts.

### 1.2. DL Applied to RA

DL and CNN have also become paramount in medical imaging, extending to the RA field. For instance, Üreten et al. proposed an automated diagnostic method using CNN to assist in diagnosing. They used 135 hand images of RA patients and healthy individuals to perform binary classification and reached 73% accuracy, with an error rate of 0.0167, thereby showing that CNN is promising in diagnosing RA [9]. Murakami et al. proposed using a conventional algorithm to segment bones as a detection method for erosion score classification using a deep CNN classifier. They used a threefold cross-validation method to evaluate the algorithm. The true and false positive rates were 78.9% and 1.43%, respectively [10], which indicated good performance. However, the algorithm could not segment intercarpal joints and often over-segmented images, which became a limitation resulting in incorrect detection. Hirano et al. proposed a CNN-based two-step method to predict JSN and erosion scores. They clipped 186 radiographs to 1860 images and 11,160 augmented images for finger joint detection using a CNN model; compared with two clinicians, the method’s was accuracy increased by 49.3–65.4% and 70.6–74.1% for JSN and erosion scoring, respectively [11]. However, they excluded a few intercarpal joints, and they mentioned insufficient training data as part of the reason for the rather low accuracy rate. They also mentioned that it was difficult to identify each area of intercarpal joints using their model. The abovementioned methods show the potential of DL in RA and the relevance of developing an intercarpal joint detection method.

Moreover, object detection technology can mark and locate the ROI from an object’s image and output the location coordinates of the object and its category. Early two-stage object detection algorithms include R-CNN [12] and its extended versions—fast R-CNN [13] and faster R-CNN [14]. YOLO and its extended versions [15,16,17,18] are well-known examples of one-stage object detection algorithms; their prominence is due to their robustness and efficiency in object detection. In addition, YOLO can predict more accurately and quickly than its counterpart DL models. With the development of DL in object detection, one-stage models have surpassed two-stage ones. For example, YOLO version 3 (YOLOv3) has higher accuracy and faster speed than faster R-CNN [16].

Recently, YOLO has achieved remarkable success in the field of object detection, having the potential to locate bone joints. Hioki et al. proposed an automatic diagnostic system for RA. They used 50 X-ray images of RA and diagnostic markers of finger joints to train and validate YOLOv3 to evaluate each of the diagnostic markers. The Sharp/van der Heijde score index of finger joint surface reduction and erosion was employed to categorize the 50 images; the prediction accuracy of each category reached approximately 80% on average, which verified the applicability of the DL technology in RA diagnosis [19]. However, they did not explicitly mention the data of commonly used evaluation indicators for object detection algorithms, such as the mean average precision (mAP). Therefore, the marking ability of this model cannot be measured, and the small dataset used may lead to insufficient data diversity.

Considering the ethics of AI, it is irresponsible to trust the AI results created by AI applications, which cannot be understood by humans. Therefore, building trustworthy AI is key to enabling AI applied in clinical practice. This ethics issue became one of the major barriers to its widespread adoption in mission-critical applications such as medical diagnosis and treatment [20]. Only 15% of the RA-related X-ray studies mentioned in the previous studies used direct explainability methods. The lack of explanation of the model results has a profound impact on the level of trust the patients or physicians have in the model. It is recommended that most of the data used by physicians for RA diagnosis should take into account the creation of deep learning models that resemble physician behavior, reducing the black-box nature of DL by simulating the behavior of medical experts [21].

After reviewing the related applications of medical AI, the literature on joint labeling in RA is inadequate to the best of our knowledge. The existing hand bone and joint labeling studies mainly focused on finger joint diagnosis; studies on intercarpal joint detection are relatively few, mainly because, compared with fingers, the bones of the wrist are more difficult for CNN detection. Therefore, in this study, we propose using YOLOv4 [15] to automatically detect RA-related phalanges and metacarpal joints to improve the existing mTSS manual workflow.

In this study, we propose an appropriate semi-automated the mTSS scoring system to speed up the workflow of mTSS while maintaining accuracy including finger joint detection and classification using an explainable deep learning method for hand X-ray images. We believe that this tremendous change will allow it to be more widely used clinically to help physicians diagnose rheumatoid arthritis.

## 2. Method and Materials

### 2.1. Image Collection and Deidentification Process

We collected medical images from Taipei Veterans General Hospital as data for this study. Deidentification in accordance with Digital Imaging and Communications in Medicine (DICOM) Supplement 142 [22], including direct and indirect personal identifiers, was processed using a self-modified DICOM Toolkit (DCMTK), 3.6.1, OFFIS, Oldenburg, Germany [23].

The RA image dataset was retrospectively collected; 400 patients older than 20 years, who met the diagnostic criteria for RA, were selected; 915 DICOM hand X-ray images comprising 12-bit grayscale images, with resolutions between 1230 × 1504 and 3015 × 2505 pixels, were obtained. The ROI labels of all images were annotated by immune rheumatologists. Each image contained an annotation set of JSN. The annotation set contained 30 classes: four proximal interphalangeal joints (PIPs), five metacarpophalangeal joints (MCPs), and six carpal bones related to the joint space measurement for each hand. Furthermore, nonrepresentative training images, e.g., broken fingers and overlapping joints, were deleted to reduce outliers to affect the training model. Two samples of included and excluded images and the three mTSS score of the images are shown in Figure 1. Finally, we randomly split the images in the dataset into 80% and 20% for detection model training and validation.

### 2.2. Definition of ROI

The number of bones in a normal hand X-ray region from fingers to wrists includes 14 phalanges, five metacarpals, and eight carpals, yielding a total of 27 bones. The left and right hands, joint/bone names, and the order of fingers are all abbreviated in the images in this dataset. R_PIP2 denotes the right hand connecting the middle phalanx of the index finger and the proximal phalanx only through the finger joints, and R_SC denotes the right scaphoid bone and head shape. The space between two bones, reclassified names, abbreviations, and full names of other abbreviations are listed in Table 1.

#### 2.2.1. The Strategy of Labeling ROI

As mentioned earlier, the annotations of mTSS ROI are susceptible to personal factors, such as past diagnosis experience and physical strength, resulting in different region sizes. In terms of vision, The JSN image feature to be evaluated is the distance between adjacent bones. Moreover, DL models require a huge amount of data for feature extraction. It is impractical in most cases to use only the 915 images we collected to complete 30 classifications of neural network models. Therefore, to make the model training practicable, we propose a method to maximize the number and accuracy of samples to predict JSN using our limited data.

In this method, the finger joints connecting finger bones and the metacarpal joints connecting the finger and metacarpal bones were considered as the same type of joint because of the small difference in their characteristics, whereas the remaining parts of the carpal joints differed significantly. To evaluate those differences, we used mTSS to reclassify 30 joint ROIs for JSN to the same category and two categories, as hand and carpal joints. Thus, for the JSN training, 915 ROIs of each of the 30 types of joints were integrated into a total of 27,450 ROIs in the first modus or 16,470 and 10,980 finger and metacarpal joint ROIs in the second modus. This method met the sample requirements and reduced the cost of model training.

#### 2.2.2. Image Preprocessing

Image preprocessing included data augmentation, feature enhancement, and image conversion. First, the interpretation of the X-ray image was adjusted via the window level (W/L) not only to enhance ROI features, such as joint and bone, but also to standardize image data from device differences and reduce the noise signals not related to the ROI, e.g., muscles. The W/L method is commonly used in DICOM medical radiographic images to interpret X-ray images in clinical practice. After several experiments and observations, we found that the W/L value of 2026/1791 and the inversion were optimal for the ROI for brightness transformation from the DICOM images. Figure 2 is an example of an image before and after applying W/L adjustment. After W/L processing, the image was downsampled from 12 to 8 bits per pixel converted to PNG format for training to reduce GPU usage to an acceptable level.

#### 2.2.3. Data Augmentation

For data augmentation, we used basic image processing methods, i.e., general shifting, displacement, brightness adjustment, and stretching. Furthermore, we used the mosaic method to increase information by mutually merging four images. The mosaic process comprised three steps: (1) reading four images; (2) performing zooming, flipping, and color gamut changes on the four images; (3) combining images and frames. Mosaics can enrich the background of the detection targets and prevent the model from relying on fixed positions for learning.

Neural networks can learn multiple images simultaneously, which increases the variability of the input image, reduces overfitting, and shortens the data processing time during batch normalization (BN); moreover, the training results can be significantly improved [15]. Figure 3 is an example of an image after using a mosaic as the input data.

#### 2.2.4. DL Detection Model

We used the YOLO series Darknet framework model, including YOLOv4, YOLOv4-tiny-3l, YOLOv4-tiny, and YOLOv3, for training, separately. These models are based on the same architecture. For example, the architecture of YOLOv4 constitutes three major parts: (1) backbone: CSPDarknet53, (2) neck: SPP + PAN, and (3) head: YOLO HEAD.

To simplify the experiment design, if an image is input to the model, it first goes through CSPDarknet53 before proposing the final three layers of convolutional data (consider a 608 × 608 image input as an example; the three layers would be 76 × 76, 38 × 38, and 19 × 19). Next, the neck part comprising SPP + PAN fuses the feature information of feature maps of different sizes. Finally, through the head part comprising BN and a 1 × 1 convolution, a prediction is made, which outputs three different feature maps. The feature map of CNN usually has a hidden layer, which is finally converted into a vector in the output layer through the fully connected layer or average polling layer. The feature map of YOLO is directly output. Different channels are usually employed to represent different features of an image. The different channels of YOLO can not only express image features, such as joints in this study, but also express coordinates and confidence. Figure 4 is the image processing pipeline for RA using YOLO series models.

#### 2.2.5. Modified YOLO Model

A crucial step is bounding box regression, which was originally proposed for R-CNN to refine the prediction results of the model [12]. Its evaluation metric is the intersection over union (IoU), which measures the degree of overlap between the bounding box and ground-truth box generated by a model. The calculation method of IoU has evolved into the following four functions [24].

IoU loss, $L_{IoU}=1-IoU\left(A,B\right)$. This is simply the difference between 1 and the intersection of prediction box *A* and real box *B*.Generalized IoU (GIoU)-loss, $L_{GIOU}=1-IOU\left(A,B\right)+\left|C-A\cup \left.B\right|/\left|\left.C\right|\right.\right..$ GIoU-loss is used to alleviate the gradient problem of the above IoU loss when the detection frame does not overlap.Distance IoU (DIoU)-loss, $L_{DIoU}=1-IOU\left(A,B\right)+\rho^{2}\left(A_{ctr},B_{ctr}\right)/c^{2}.$ DIoU-loss is a simple penalty added to minimize the standardized distance between the center points of the two detection frames, which can accelerate the convergence process of the loss.Completed IoU (CIoU)-loss, $L_{CIoU}=1-IOU\left(A,B\right)+\rho^{2}\left(A_{ctr},B_{ctr}\right)/c^{2}+\alpha \cdot \nu$. Compared with DIoU-loss, CIoU-loss considers the aspect ratio information.

In detecting human-made objects, such as automobiles, the aspect ratio is helpful for model evaluation; however, for the joints in this study, the annotations impact the individual differences of different immune rheumatologists. The joint itself also requires different lengths and widths for annotation due to the angle; thus, CIoU-loss is unsuitable for evaluating the proposed model. We decided to use DIoU-loss and DIoU-nonmaximal suppression (NMS) to eliminate redundant detection frames instead of CIoU-loss and the conventional greedy NMS. We also set the average accuracy rate of all category mAPs to evaluate the results when the IoU threshold was 0.5 and adjusted the input data resolution after selecting the best model to achieve an optimal result.

### 2.3. Joint Classification

We used the results of the detection model as the input of the classification model. In this study, the detection model and the classification model were trained at the same time, and we simulated the results of the detection model by cropping the joints of the image adjusted by Window level according to ROIs. According to the labels provided by immuno-rheumatologists, these images were scored from 0 to 4, depending on the severity of the mTSS-based disease. If the size of these selected areas was less than 128 × 128, the selected area was expanded to 128 × 128 to fit the size of the input shape. After the image was cropped, it was divided into healthy, mild, and severe categories corresponding to mTSS scores of 0, 1–2, and 3–4. In total, 21,031, 539, and 1406 images were cropped according to the classification category. In order to balance the training data, we only used undersampling for those samples that were classified as healthy to balance the data while retaining more diversity in the seriously ill samples. The completed data were divided into 837, 521, and 1371 pictures respectively, and an 80:20 split was used for training and validation, respectively.

#### 2.3.1. DL Classification Model

In the training process, we used EfficientNet as the backbone, adjusting input samples larger than 192 × 192 to 192 × 192 as training input. The depth and width of the model of the EfficientNet were scaled by the size of the input image according to a fixed ratio, i.e., compound model scaling, and the neural architecture search was used to generate the baseline model, i.e., EfficientNet-B0 (B for baseline, 0 for *n* = 0) [25]. On the basis of this model, seven expansion models were generated from *n* = 1 to 7 for different input data resolutions. This allowed the model size to be allocated in the most reasonable way to reduce the training time. Since the input data of the classification models were cut from the results of the localization model, their resolutions were relatively small. Therefore, we used EfficientNet-B1 as the classification model. Since we used an 80:20 data split, we applied fivefold cross-validation to reduce the bias of the model.

To improve the accuracy of the classification model, we used the Adam optimizer to set the initial learning rate to 0.001 to determine the optimum value during the training process. Then, we used the verification set to evaluate the model at the end of each epoch. If the verification loss was not improved after four consecutive cycles, the learning rate was reduced by 0.1 times. The batch size during training was 16. To reduce the impact of data imbalance on model training, the class weights of the classification were set to 2.63, 1.63, and 1, respectively. We also utilized early stopping to avoid overfitting, which terminated the training process after five consecutive epochs if the validation loss did not improve.

#### 2.3.2. Explainable Models

In addition to the accuracy of the mTSS classification model we proposed, it is very important for clinical decision making to illustrate how to explain the plausibility of our proposed model if it may be used in clinical practice in the future. Therefore, our proposed model was evaluated using the gradient weighted class activation mapping (Grad-CAM) to visualize the heatmap of the predicted results in the joint area. Grad-CAM uses the gradient information returned from the last convolutional layer in the CNN model to determine the importance of each neuron to a specific decision value of interest. For each input image, Grad-CAM eventually generates a heat map indicating the areas that the model considers important [26].

## 3. Results

### 3.1. Joint Localization

To compare the training results of the four YOLO models, which were trained using the same parameters (Table 2), all models were not modified by changing the IoU-loss function and NMS to DIoU-loss and DIoU-NMS, respectively. The Darknet framework was used to train the models to detect and locate the JSN ROIs from X-ray hand images. The training experiment involved two steps. Figure 5 shows the training strategy of this study.

**Experiment** **1.**The performances of the four YOLO series models and their training results on two datasets were compared. We also trained two non-YOLO series models (EfficientDet-D0, faster R-CNN) on these two databases to verify the performance of the YOLO series models. The first dataset labeled all hand joints as a single class. The second dataset labeled the hand joints as finger and wrist joint classes.

**Experiment** **2.**The performances of the YOLO model trained with different resolutions of RA images were compared. The resolutions used in this step were initially 416 × 416, then 320 × 320 and 256 × 256, and finally 608 × 608. All experiments were compared on the same test set; the results were calculated using mAP with the IoU threshold set to 0.5 (mAP@0.5).

#### 3.1.1. Comparison of Training Results

As shown in Table 3. The best training result of experiment 1 was achieved by training YOLOv4 with the first dataset, which labeled all hand joints as a single class. The mAP@0.5 of the model reached 0.83 for JSN ROIs. All models, performed better when trained on the first dataset than on the second dataset. When trained on the first dataset in the YOLO series models, YOLOv3 achieved slightly worse performance than YOLOv4, whereas YOLOv4-tiny performed the worst among all models. The experimental results also confirmed that the YOLO series models were superior to other two-stage models in terms of overall performance and training time.

The training results of experiment 2 are shown in Table 4. After increasing the input image resolution to 608 × 608, the model’s mAP@0.5 improved to 0.71, which was slightly higher than the original resolution (416 × 416), whereas the model’s mAP@0.5 decreased to 0.68 when the resolution decreased to 256 × 256 and 320 × 320.

#### 3.1.2. Model Testing and Analysis

The performance results of the proposed modified YOLOv4 model and the conventional YOLOv4 model on the test set are shown in Table 5, the final modified model had an mAP@0.5 of 0.92; the precision, recall, and F1 score reached 0.95, 0.94, and 0.94, respectively.

The detection results of the modified model on the test images are shown in Figure 6. The image contained two hands with different symptoms for demonstration. The model could fully detect 15 JSN ROIs in the X-ray image of mild RA. Although the hand joints in the X-ray image of severe RA were blurry, the model could still fully detect 15 JSN ROIs.

### 3.2. Joint Classification

The classification results and confusion matrix are presented in Table 6. Among the 545 test data, the precision of severe, mild, and healthy reached 0.91, 0.79, and 0.9, respectively. We also tried to explain the plausibility of the classification model to validate the AI results in the joint area that matched the diagnosis by two rheumatologists. The explanation of the model is shown in Figure 7.

## 4. Discussion

In our literature review, we found that many scholars encountered the problem of an insufficient sample size [7,8,10,11,19]. Our method explored the feasibility by standardizing data annotation files to reduce the computing power requirements for the CNN model and improve the positioning accuracy with fewer than 1000 images. In this study, YOLOv3 and YOLOv4 models were used to detect joints with RA from hand X-ray images. Both models are improved versions of the conventional YOLO model.

In Experiment 1, we found that, if YOLO models were trained with the second dataset, the average precision of detecting wrist joints was significantly lower than that of detecting finger joints, indicating that the reason for these low mAPs was that these models could not effectively learn the features of wrist joints individually; therefore, models performed better when trained on a single class.

In Experiment 2, the results showed that the increase in image resolution improved the model’s labeling ability. Owing to the limitations of hardware used and our strategy on model workflow in this study, the resolution of the input image could not be further increased. Training with higher resolution can also improve the model’s detection and classification ability. When the input image resolution was 608 × 608, the mAP@0.5 score reached 0.83, and the precision, recall, and F1 scores reached 0.87, showing that the proposed model could achieve excellent labeling capabilities in different RA severities.

In radiographic diagnosis for RA with the mTSS, the standard deviation of the mean mild score is large; thus, it is difficult to determine a precise score by a human vision. The result proves the importance of the proposed classification model in assisting physicians to screen the RA severity in a very short time. Even though, with the dynamic adjustment of the learning rate, the accuracy of the training set reached 1, the accuracy of the validation set still has room to improve. The proposed model was effective during testing, although the classification of mild diseases was not more accurate than the other two categories due to the limited amount of mild data. Even though the average accuracy achieved was 0.88, the proposed model is still better than the other traditional models. We do believe that increasing the resolution of the input data, as well as adding more samples of diseased joints, can further improve the training results.

In addition, the clinical experts in our research group proposed that, in the imaging examination protocol, the two-hand X-ray image is the most common imaging protocol for saving imaging time and reducing the X-ray exposure for patients. However, using the two-hand X-ray imaging protocol could result in shadows due to overlaying by other bones occurring in joint spaces as a function of the large angle of projection from the X-ray emission center. When using a one-hand X-ray, the shadows in joint spaces are minimized because the projection angle is smaller. Figure 8 shows a comparison between the different projection angles from two-hand and one-hand X-ray imaging protocols.

We suggest that applying the one-hand X-ray imaging protocol can improve the accuracy of the mTSS classification model in determining mild disease if it is used in clinical practice. According to our experiment, we discovered that the shadow caused by bones is a key factor influencing the classification model when fewer samples are used. Thus, our hypothesis, which was verified by physicians in our team, was that this might have a negative impact on the deep learning model. Thus, to verify our hypothesis, we extracted five two-hand and 10 one-hand images from the test dataset to test the model, and their accuracies were 0.87 and 0.94, respectively. According to this verification, the accuracy of the one-hand X-ray image was higher than that of the two-hand X-ray image.

The feasibility and effectiveness of the proposed YOLO-based model were verified to detect and locate the area of joint space narrowing in X-ray hand images. Since the data provided by the localization model are stable, we developed the classification model on the basis of the localization results to develop the mTSS hand computer-aided diagnosis system, e.g., to assist physicians to screen RA severities in a very short time. We are still improving the detection capability of the models, expanding their application range, and developing a complete set of CAD systems covering hands, feet, and knees. This has become the most complete mTSS hand computer-aided diagnosis system with explainable results to the best of our knowledge.

In RA clinical practice, the imaging-related scoring process is a time-consuming and labor-intensive process; adopting DL and machine learning (ML) can reduce the barrier to its adoption into daily clinical practice, especially for screening, as well as improve the efficiency of data collection for research purposes. Adopting image preprocessing to enhance the features for ROI via the W/L method and brightness transformation from the DICOM images can help improve the model accuracy by not only enhancing the specified features of the radiographic image but also reducing the noise signals. The technical concept of our proposed mTSS classification model can be applied to similar RA diseases to achieve the widespread use of quantitative image analysis by DL and machine learning in RA clinical practice. For example, for the application of systemic sclerosis (SSc), an automatic tool for the Warrick score (WS) can be developed to define the clinical outcome by defining the severity of pulmonary invasion in high-resolution computed tomography (HRCT). Those results with explainable AI can also help physicians achieve early prevention, as well as identify the WS severity level to screen SSc severities in a very short time. In this way, it will be possible to combine clinical and instrumental data to assist immuno-rheumatologists to predict early lung involvement for early diagnosis in SSc [27].

## 5. Conclusions

In this study, we not only improved the mTSS manual labeling process using state-of-the-art object detection technology to automatically detect and locate the regions of joint space from X-ray hand images, but also provided a joint classification model depending on the severity of the mTSS-based disease. The contributions of this study are summarized below.

DL was used to establish a state-of-the-art model for automatically locating all hand joints of ROIs of JSN in “mTSS”.CIoU-loss was converted to the previous version of DIoU-loss when facing non-artificial objects and other materials where the aspect ratio does not need to be considered, thus helping improve the model accuracy.Experimental results showed that, considering the human visual ability to identify data quantity or quality, training the models on a dataset that labeled all hand joints as a single class yielded better performance than training them on a dataset that labeled hand joints as finger and wrist joint classes.DL was used to propose an efficient mTSS-based classification model, explaining the plausibility of using Grad-CAM to validate the AI results in the joint area, which can assist physicians to screen RA severities in a very short time.

## Figures and Tables

**Figure 1 biomedicines-10-01355-f001:**
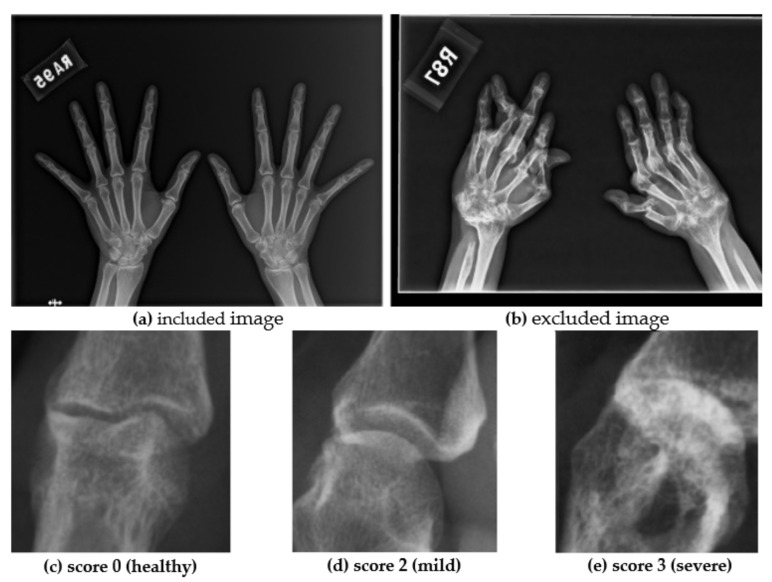
Comparison of examples of normal X-rays and RA images: (**a**) normal image; (**b**) image filtered out because of distortion; (**c**) healthy joint; (**d**) mild joint; (**e**) severe joint.

**Figure 2 biomedicines-10-01355-f002:**
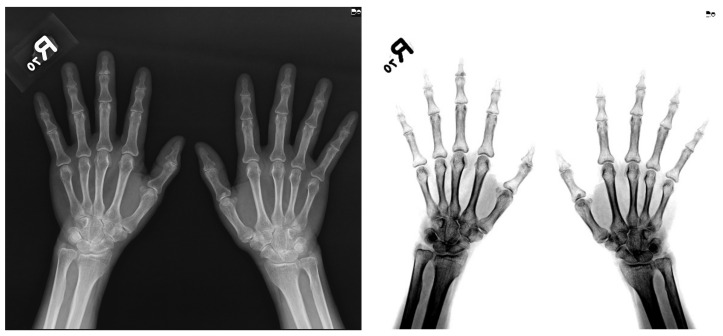
A sample after window level adjustment.

**Figure 3 biomedicines-10-01355-f003:**
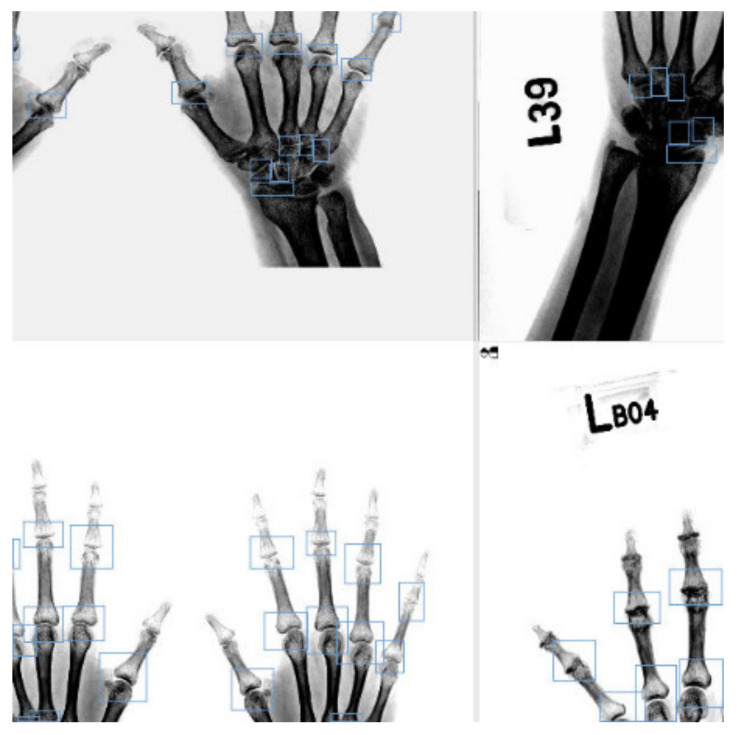
Example of an image after using a mosaic as the input data.

**Figure 4 biomedicines-10-01355-f004:**
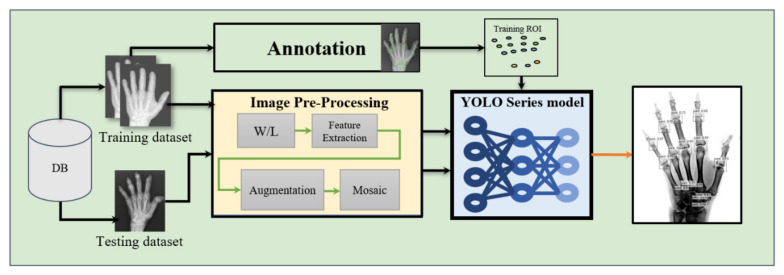
The image processing pipeline for RA using YOLO series models.

**Figure 5 biomedicines-10-01355-f005:**
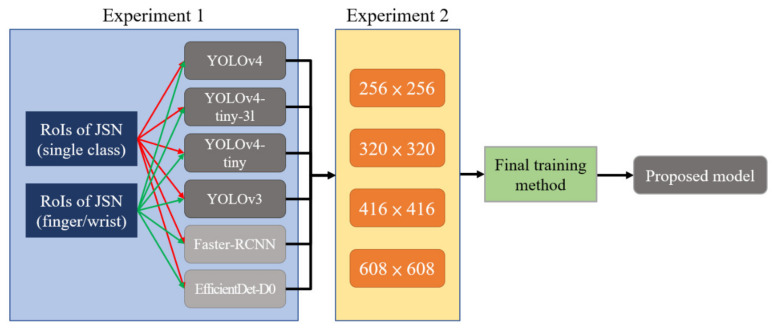
training strategy of detection models in this study.

**Figure 6 biomedicines-10-01355-f006:**
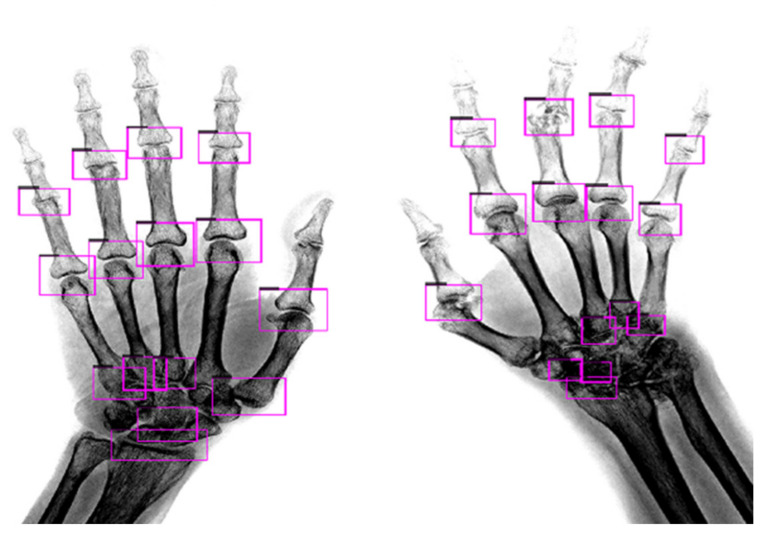
JSN model detection results; the left side of the image shows a hand with mild RA, while the right side of the image shows a hand with severe RA.

**Figure 7 biomedicines-10-01355-f007:**
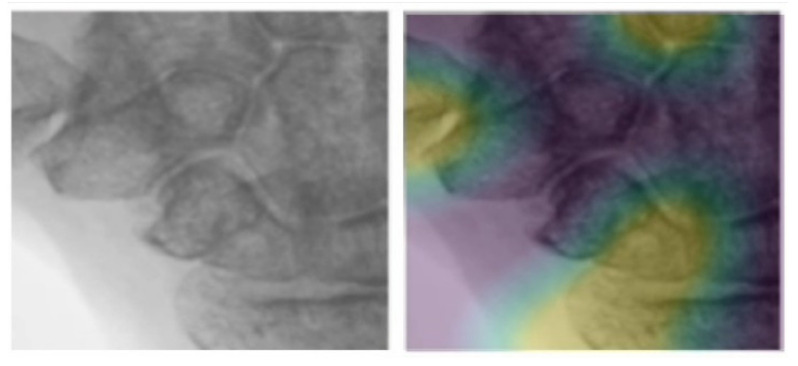
Explanation of the model compared with the heatmap for the classification results in a wrist joint area.

**Figure 8 biomedicines-10-01355-f008:**
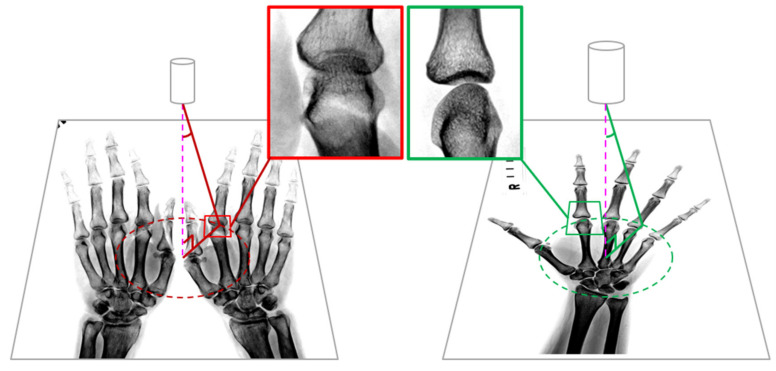
Comparison between the different projection angles from two-hand and one-hand X-ray imaging protocols: (left) two-hand X-ray; (right) one-hand X-ray.

**Table 1 biomedicines-10-01355-t001:** the abbreviations of the ROIs.

Training as 1 Class	2 Classes	Simplification	Full Name
ROIs	Finger	PIP	Proximal interphalangeal joint
		MCP	Metacarpophalangeal joint
	Wrist	CMC	Carpometacarpal joint
		TPM	Trapezium
		SCP	Scaphoid
		LUN	Lunate
		RAD	Radius
		UNA	Ulnar
		SC	Scaphoid–capitate joint
		SR	Scaphoid–radius joint
		ST	Scaphoid–trapezium joint

**Table 2 biomedicines-10-01355-t002:** Parameters of YOLO model training.

Parameters	Value	Parameters	Value
Batch	64	Decay	0.0005
Subdivisions	32	Learning rate	0.001
Width *	416	Max batches	6000
Height *	416	Policy	4800, 5400
Momentum	0.949	Scales	0.1, 0.1

* In example of 416 × 416 resolution.

**Table 3 biomedicines-10-01355-t003:** Result after first step, showing that single-class methods had better results.

	YOLO4	YOLOv3	YOLOv4-tiny-3l	YOLOv4-tiny	Faster-RCNN	EfficientDet-D0
Dataset I	0.71	0.66	0.65	0.61	0.65	0.63
Dataset II	0.63	0.58	0.61	0.55	0.59	0.58

**Table 4 biomedicines-10-01355-t004:** Training results of step 2 (calculated by mAP@0.5).

Model	Resolution	JSN mAP@0.5
YOLOv4	256 × 256	0.68
	320 × 320	0.68
	416 × 416	0.70
	608 × 608	0.71

**Table 5 biomedicines-10-01355-t005:** Evaluation of research models YOLOv4 (608 × 608).

Evaluation	JSN				
YOLO v4 Model	Original	With WL	With WL + Mosaic	With WL+ IoU and NMS	Proposed Method
mAP@0.5	0.71	0.75	0.78	0.8	0.92
Precision	0.67	0.72	0.76	0.77	0.95
Recall	0.86	0.88	0.91	0.92	0.94
F1-Score	0.75	0.79	0.83	0.84	0.94

**Table 6 biomedicines-10-01355-t006:** Results of classification model EfficientNet-B1 (192 × 192).

Evaluation	Precision	Recall	F1-Score	Support
Healthy	0.91	0.98	0.94	166
Mild	0.79	0.72	0.75	103
Severe	0.9	0.89	0.89	276
Accuracy	-	-	0.88	545
Macro avg	0.87	0.86	0.86	545
Weighted avg	0.88	0.88	0.88	545

## Data Availability

The experimental datasets used in this manuscript are not public available due to ethical reasons. The detailed records of our proposed model are public available on https://github.com/NicholaxKAmIL/RA_PaperData (accessed on 5 June 2022).

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
