# Peer review of "Deep Learning-Based Computer-Aided Diagnosis of Rheumatoid Arthritis with Hand X-ray Images Conforming to Modified Total Sharp/van der Heijde Score"

_biomedicines, 2022, doi:10.3390/biomedicines10061355_

Round 1

Reviewer 1 Report

The paper is very interesting and well written. The application od learning-based computer technique appears useful to improve the methodology and criteria for AR diagnosis. These findings confirm the importance of apllication of novel techniques in medicine. I suggest to briefly discuss in the discussion for example the novel application of machine learning in systemic sclerosis (see and add as references papers by Murdaca et al concerning this topic) opening the possibility to apply this statistical methodology in AR studies.

Reviewer 2 Report

In this paper, the problem of RA diagnosis via hand X-ray image analysis is studied.
For this purpose, four YOLO variants are applied.

After studying the manuscript and the related references the following comments are stated:

1) The technical presentation is clear and the manuscript structure appropriate.
2) The reviewer encourages the authors to provide the used image data in an open-access framework to the research community in order to help the evolution of this research field.
3) The main weakness of this work is the absence of any comparative study with similar methods already proposed in the literature.

Round 2

Reviewer 2 Report

The authors have addressed all the reviewer's concerns. Thank you for your efforts.